# Effect of Viscosity and Air Gap within the Spinneret on the Morphology and Mechanical Properties of Hollow-Fiber Polymer Membranes for Separation Performance

**DOI:** 10.3390/polym16142090

**Published:** 2024-07-22

**Authors:** Sirisak Seansukato, Sathish Kumar Ramachandran, Sivamesh Lamlong, Wirach Taweepreda, Gangasalam Arthanareeswaran

**Affiliations:** 1Polymer Science Program, Division of Physical Science, Faculty of Science, Prince of Songkla University, Hat-Yai 90110, Songkhla, Thailand; seansukato@gmail.com (S.S.); rsathish1989@gmail.com (S.K.R.); 2Saveetha Institute of Medical and Technical Sciences (SIMATS), Chennai 600 077, Tamil Nadu, India; 3Faculty of Environmental Management (FEM), Prince of Songkla University (PSU), Hat Yai 90110, Songkhla, Thailand; sivamesh.la@gmail.com; 4Membrane Research Laboratory, Department of Chemical Engineering, National Institute of Technology, Tiruchirappalli 620 015, Tamil Nadu, India; arthanareeg@nitt.edu

**Keywords:** PVC, PEG, hollow fiber membranes, spinneret

## Abstract

Hollow-fiber membranes for nanofiltration were prepared from the blending of Poly (ethylene glycol) (PEG) with Poly (vinyl chloride) (PVC) with different PEG molecular weights (400 and 4000 g/mol) and PVC via a dry/wet spinning process. In the spinning process, the effects of air gap, wind-up speed, dope extrusion rate, and bore extrusion rate were examined. In addition, the different lengths of the center tube, which acted as the inner-side fiber diameter during the preparation of hollow-fiber membranes, were studied. This research was investigated in order to observe the morphological, dielectric, and dynamic mechanical thermal properties to identify a suitable preparation of a hollow-fiber membrane for feasible applications. The morphology of the PVC-580 blended PEG-400 5 weight percent hollow-fiber membrane was seen to have a dense skin on both the inner and outer fiber surface, along with a suitable dope viscosity. Moreover, it offered finger-like substructures that could provide a high applicable feed-stream permeability and selectivity. Finger-like substructures were present on the near inner fiber surface at the controlled center-tube length of 0.3 cm, more so than at the center tube of 1 cm. This was because the solvent and non-solvent in the lumen tube exchanged more quickly than they did in the coagulant bath. The effect of the wind-up speed during the spinning process was significantly influenced by an affordable hollow fiber that can be indicated by the drawing ratio (*λ*). It was found that the drawing ratio of 3.3 showed a thickness thinner than 2.6 and 2.0, respectively. In summary, a controlled wind-up speed, an acceptable dope viscosity, and—most importantly—an agglomerated time resulted in membrane preparation.

## 1. Introduction

In the last 40 years, hollow-fiber membranes have been a source of interest as a favorable apparatus for membrane separation because of their high separation efficiency, self-mechanical support, good flexibility, and easy handling during fabrication [1]. However, the preparation of hollow-fiber membranes consisting of permeability and selectivity often encounters challenges due to the complexity of the spinning process. A major problem for membrane scientists is how to fabricate hollow-fiber membranes with an adapted structure to obtain a high pressure application, plasticization, and suitable morphology. A defect-free hollow-fiber membrane and an ultra-thin dense-selective layer are the most desirable configuration because these geometrics lead to a great increase in permeability. An ultra-thin dense-selective layer can reduce transport resistance and lead to enhanced selectivity. In addition, the sub-layer within the membrane seems to cause a plasticization effect that can lower the membrane’s performance [2]. As a consequence, the preparation of the hollow-fiber membrane must consider many factors that are involved in processes in order to meet requirements.

Hollow-fiber membranes were first presented in a series of patents by Mahon in 1966, assigned to the Dow Chemical Company [3]. Recently, hollow fiber has been used in a large number of commercial applications such as medical devices, water reclamation, pervaporation, and gas filtration due to its good mass transfer properties and high ratio of membrane area per volume [4]. Many different methods to prepare synthetic membranes have been used, including sintering, track-etching, phase inversion, and coating. Among the various methods, phase inversion is simply applied in the fabrication of available membranes. The phase inversion method is divided into four broad categories [5]: the wet/wet phase inversion, the dry/wet phase inversion, the dry/dry phase inversion, and the dry/dry–wet phase inversion. The difference between dry and wet processes is the use of the outlet of the spinneret or casting knife, which exposes the dope solution from a closed coagulant bath and exposes it to air or immerses it directly in a non-solvent coagulation bath to be used in the precipitation process. Accordingly, those methods can be used for a phase inversion process. Moreover, during the process of exchanging a solvent and a non-solvent, the evaporation of the solvent, forming by thermal precipitation, and cooling by immersion in a non-solvent bath can occur [6]. Normally, the principal factors during the spinning process are the dope composition, dope viscosity, temperature, dope pumping rates, bore solution, bore pumping rates, coagulation temperature, and fiber drawing rates [7].

In general, polymeric hollow-fiber membranes have been prepared by the favored dry/wet spinning process using tube-in-orifice spinnerets [4]. During this process, the polymer solution or the dope solution is extruded from the outer orifice of the spinneret and the non-solvent or bore solution is extruded from the center tube of the spinneret, respectively. Usually, a dense-selective layer or defect-free skin layer in asymmetric hollow-fiber membranes may result from the packing of kinked polymer chains [7,8] that gain influence from the controlled parameter during the spinning process. In addition, the applications of hollow-fiber membranes are considered depending on the direction of the feed flow, permeate flow, and retentate flow, which are vital factors in the use of membranes in order to obtain optimum performance. Poly (vinyl chloride) (PVC) was selected for membrane fabrication. It displays behavior as a semi-glassy polymer type, and is often preferred as a membrane material because it is highly durable and easy to fabricate in many aspects [9,10]. PVC can be prepared as a dense membrane material probably because of its suitable properties that consist of its low cost, excellent chemical and physical properties, and acid and alkaline resistance, as well as mechanical properties [11,12,13,14,15,16]. In addition, PVC has many advantages over other types of plastic such as Polyethylene (PE), Polypropylene (PP), and Polystyrene (PS), as it can be improved for various property requirements (soft and hard) and can also be recycled because of its thermoplastic type. Thus, it is able to save a lot of energy and reduce the amount of plastic waste [17]. The addition of Polyethylene glycol (PEG) content is a remarkable method that is often used as an additive to increase the permeability and porosity of membranes. PEG is also known as a phase change material (PCM) [18,19] because it comprises a molecular structure indicated as H–(O–CH_2_–CH_2_)_n_–OH which is formed as a liquid or a low melting wax, like a solid [20]. The repeating unit of PEG’s structure contains the O atoms in ethylene oxide units which create the hydrogen bonds (H-bonds) with the H atoms in the CH–Cl group of PVC’s back bone [21]. Sadeghi et al. [20] reported the effect of the addition of the different molecular weights of PEG on the effectiveness of gas permeability and selectivity. In addition, the influence of the concentrated polymer solution with additives between PVC and PEG on their miscibility was investigated by Castro and Neiro [22,23]; they explained that the electron-rich O atoms in PEO may act as donors and have attractive interactions with the electron-deficient Cl atoms in the PVC structure. Surprisingly, numerous researchers have employed readily available experiments to examine the impact of varying PEG molecular weights on improving membrane performance. Kim and Lee [24] reported that, when the molecular weight of the PEG additive was added from 600 to 2000 and 6000 Da in the polysulfone matrix, the pure water flux increased and solute rejection decreased; meanwhile, with an enhancement in the ratio of PEG-600 contents to the solvent, the water flux increased, and solute rejection decreased. Furthermore, PEG-400, PEG-6000, and PEG-20000 were utilized as additives in the polysulfone membrane qualities described by Chakrabarty and colleagues [25]. They found that the porosity remained almost invariable in terms of morphology when increasing the PEG molecular weight from 400 to 6000 Da; while, at the same time, increasing with a PEG molecular weight from 6000 to 20,000 Da. The increases in the PEG molecular weight are influenced by the poor miscibility of the dope solution. The viscosity of the dope solution is a main parameter that mostly regulates the morphology, shape, and permeability of hollow-fiber membranes. Varying the viscosity of the dope solution can influence the flow rate, the draw ratio, and the spinning speed of the hollow fiber [26]. It is well-known that hollow-fiber membranes have been significantly prepared by polymer materials via a wet spinning process. In the typical concentrated solution spinning and phase inversion method, a hollow-fiber membrane is fabricated by an extruding capillary spinneret along with wind-up stretching fiber, whereas a non-solvent solution acting as the bore liquid is extruded through the internal capillary. Aggregation and the creation of the polymer structure during the processes require the use of both bore liquid and an external coagulant [27,28,29,30,31,32]. The diameter and wall thickness of the fibers are controlled by the inner/outer diameter capillary which is quite limited in the changing capillary diameters.

According to the capillary diameters, these factors can result in a different efficiency separation. Significantly, it is a practical challenge to adapt the size of the hollow-fiber membranes by using a blunt needle. Yao and his colleagues [33] have previously reported that they could create ultrafine hollow-fiber membranes using a single-orifice spinneret, which was then extruded using a blunt needle. It was found that this method is much simpler than the conventional spinning process.

## 2. Materials and Methods

### 2.1. Membrane Materials and Chemicals

Polyvinyl chloride (PVC) was kindly supplied by Thai Plastic and Chemicals Co., Ltd., Rayong, Thailand, *K*-value 580, 610, and 710. The solvent *N*,*N*-dimethylformamide (DMF 99.8%) was purchased from Ajax Finechem Pty Ltd., Sydney, Australia. Poly(ethyl glycol) (PEG 400 and 4000 Dalton) was purchased from Honam Petrochemical Corp., Seoul, South Korea. Dope solution was prepared by dissolving PEG in DMF and stirred for several minutes. Then, PVC was subsequently added with continuous stirring and heating at 60 °C until the solution is completely dissolved and homogeneous. The resultant dope solution was continuously kept at 60 °C before spinning process.

### 2.2. Dope Viscosity Measurement

The dope temperature was kept at roughly 60 °C to prepare spinning solutions until every polymer was examined. The homogeneous solution of PVC and PVC-blended PEG was measured using a Brookfield Viscometer (Model 5XHBTCP, Middleboro, MA, USA). The experiments were carried out using spindle number 4. The flux times were recorded every 30 s. Measurements were performed at a constant temperature at 25 °C.

### 2.3. Fabrication of PVC–PEG Hollow-Fiber Membrane

The mixed solution or dope solution of PVC and PEG was homogenized by using hot plate magnetic stirrer of VS-130SH from Vision Scientific Co., Ltd. (Daejeon-Si, Korea). The mixed solution can be completely dissolved in homogeneous solution at temperature 60 °C for approximately 24 h. The detailed conditions of hollow-fiber spinning are presented in Table 1. Then, mixed solution was poured into in situ extruder which was already installed with blunt needle. The combined solution, which is insoluble in water, is forced into the coagulant bath and toward the fiber to wind-up drum during the spinning, phase separation, and stretching processes. The overhead stirrer IKA RW 20 digitals regulate the collective speed of this movement. Then, the fibers are immersed in water at 25 °C for 2 days to ensure the releasing of the solvent. The hollow-fiber membrane was dried at 30 °C. The fabrication process of hollow-fiber membrane is shown in Figure 1.

It was discovered that, during the spinning process, the draw ratio and shear rate were what caused their selectivity and permeability. It is explained that the uncoiling motif of molecular chains is stimulated by the elongation field, which leads to their compressed packing in the skin layer. In addition, the effect of draw ratio is affected to improving the membrane structure during spinning process due to the elongation-arranged molecular orientation which accelerates polymer chains to provide more regular packing and more mono-disperse space. Moreover, it is believed that the increasing air gap distance and wind-up speed may play an important role in membrane morphology due to the fact that the elongation stress is eliminated, and the retarded chain relaxation and orientation. The drawing ratio (*λ*) during the spinning process is presented as follows [34,35]:(1)$$\lambda =\frac{V}{V_{0}}$$
where $V_{0}$ is the extrusion speed and V is the wind-up speed.

### 2.4. Morphological Analysis

The geometrical characteristics and morphology of polymeric membranes were studied using scanning electron microscope (SEM). The samples of membranes were frozen by liquid nitrogen before testing in order to obtain broken samples, and then mounted on sample stubs with double-sided tape. The preparing samples were sputtered with thin layer of gold. The study of morphology with SEM can be observed in cross-section, surface, and pore size of membranes. The investigated morphology of the prepared membranes was observed with scanning electron microscopy, Quanta-5800 (Thermo Fisher Scientific, Waltham, MA, USA). The surface and cross-section can be shown from different aspects as a result of different components in dope solution and fabricated membranes.

### 2.5. Contact Angle Measurement

The contact angle (*θ*) is the angle at which a liquid/vapor interface meets a certain solid surface. Membranes were determined by the hydrophobic and hydrophilic property, when the molecules of a liquid are strongly attracted to the membrane surface as a result of a drop of liquid being completely spread out on membrane surface, corresponding to a contact angle of 0°. While the membrane surface is hydrophobic, the contact angle is 90°. Therefore, the morphology of membranes can indicate that the hydrophobic property is *θ* > 90° and hydrophilic *θ* < 90°.

### 2.6. Dielectric Properties’ Measurements

According to improving PVC–HF membrane with additive PEG, the blended membranes have defect structure and provide free volume. Moreover, they still contained an O-H group as functional group of PEG backbone structure [36]. Consequently, the comparison of the dielectric properties of the PVC—pure and PVC-blended PEG membrane in various contents can be provided by alternative indication on their structural polarity of the polymer materials. Normally, the polymer materials have a high dielectric constant with which they are able to highly retain a charge, and are not a good conductivity of electricity. It is evident that, due to important differences in charge mobility across layers of distinct phases, alternative accumulation and dissipation of the charge occurred at these interfaces under oscillation of the electric field [37]. Dielectric measurements were carried out in the frequency range 1 MHz up to 300 MHz by using precision LCR meter (Agilent 4285A, Santa Clara, CA, USA).

### 2.7. Dynamic Mechanical Thermal Analysis Measurements

The dynamic mechanical properties of the fabricated membrane were measured using DMTA (dynamic mechanical thermal analyzer, Model-V, supplied by Rheometric Scientific, Reichelsheim, Germany). The cantilever mode of the deformed sample was carried out under the test temperature range from 0 °C to 500 °C, the test frequency being 1 Hz. The cooling process was achieved through liquid nitrogen. The terms of this method were presented in terms of loss tangent $\left(\tan\delta \right)$ and glass transition temperature $\left(T_{g}\right)$. In this investigates, loss tangent was the ratio of loss modulus $\left(E^{\prime \prime }\right)$ to storage modulus $\left(E^{\prime }\right)$; meanwhile, $T_{g}$ was perceived from the loss modulus peak.

### 2.8. Membrane Flux Performance

Membrane permeate flux (i.e., volumetric flux of water) was determined at predetermined time intervals by measuring the weight changes in the feed tank with a digital mass balance connected to a computer data logging system. A hollow-fiber membrane module was prepared with an effective membrane area of around 7.2 cm^2^ and a length of around 20.0 cm. A standardized digital conductivity meter (UTV, US) was used to measure the salt concentrations in the feed and product water for determining membrane selectivity. The volume of permeate collected was used to describe flux in terms of liter per square meter of active membrane area per hour (L·m^−2^·h^−1^).

## 3. Results and Discussions

### 3.1. Viscosity of Dope Solution

The rheological behavior of polymer dope solutions is another vital parameter that is involved in elongation flow during the spinning process. The results of this effect must be considered because the elongation viscosity of the dope solution is induced by the gravity of the nascent fibers in the air gap distance [34]. The viscosity measurement is a vital method because of its simplicity and reliability [38]. Theoretically, the absorptive interaction among the chain mobility of polymers in solution may induce the expanding of the high viscosity caused by the shrinking of the chain mobility, similar to the tendency in Figure 2a, which shows the viscosity of the three *K*-value of PVC 14% in DMF without PEG concentrations resulting from the increase in different degrees of polymerization. It is observed that different *K*-values have a significant influence on the viscosity of the spinning solution resulting in different morphological efficiencies. The effect of the PEG additive on the dope rheology was considered to be significant because the viscosity of the dope solution is an important parameter relating to the kinetics of the phase inversion in membrane fabrication. In Figure 2b, given the tendency of viscosity containing PVC 580 blended with 5 wt%, 10 wt%, and 15 wt% of PEG-400 and PEG-4000, both different PEG contents and PEG molecular weights present distinctive effects on the dope viscosity. Longer PEG chain segments for the PEG with a higher molecular weight may grow into a denser macromolecule chain entanglement, which may be chargeable for the viscosity increase [39]. In contrast, the minimized viscosity may be due to the relatively poor interaction between PEG and DMF resulting in a viscosity decrease. Significantly, the segments of PEG chains tend to attract each other in a poor solvent, because DMF is a poor solvent of PEG, resulting in the shrinkage of PEG chain segments participating in single-polymer coils [40]. Aside from the aforementioned situation, since the viscosity of these dope solutions from different *K*-values and PEG molecular weights are low, it seems that the increasing of phase inversion due to rapid liquid–liquid demixing overcomes the kinetic effect [30]. This effect resulted in large finger-like structures and induced the membranes to have an open structure that can be expected to increase gas permeability because their structure contains a high porosity within the bulk structure [41].

### 3.2. Morphology of PVC Fiber Membranes with Unemployed Center Tube and Controlled Air Gap 1.0 cm

In this preliminary experiment regarding the hollow-fiber-spinning process, a variety of dense PVC-based membranes were prepared via a phase inversion method, then fabricated by a dry/wet spinning process without bore fluid, respectively. Significantly, the beginning of the fabrication is dope spinning by in situ spinning with an unemployed metal needle to study the basic morphology of the fiber membrane with the controlling air gap distance at 1 cm and wind-up speed at 50 rpm. The cross-section, cut-cleaved section, outer surface, and skin layer were examined through SEM (Figure 3). A neat PVC membrane (P-0) was prepared by the lowest-viscosity solution, without blending PEG which plays a vital role in improving membrane efficiency. In this sample, the exchanging of the solvent (DMF in the dope solution) and non-solvent (water in the coagulant bath) rapidly transferred as well because of its high soluble efficiency between DMF and water; this occurrence promotes nuclei growth due to its solvent power decreasing, resulting in the chain entanglement of PVC molecules tending to align themselves packed less closely [30]. Therefore, the medium finger-like structures are obtained. For the mixing of different molecular weights and ratios of PEG, the P-G4-5 and P-G4-10 morphology causes an increase in the length of the finger-like structure, obliviously presented in P-G4-5, P-G4-10, P-G4K-5, and P-G4K-10. To better understand the formed behavior, previous research has explained the vicious fingering concept which has been used for studying the formation of the finger-like, sponge-like, and macro-void structure. Specifically, the behaviors of PEG with an increase in PEG molecular weight showed more holes—in other words, a macro-void structure—within the substructure because the high mass might be trapped inside the dope during the phase inversion process. While the higher PEG concentration can be explained by when the dope solution made contact with water in the coagulant bath, the instantaneous de-mixing occurs at the skin surface which easily dissolves PEGs in water due to their good solubility [42]. Thus, these trapped PEGs were expected to be dissolved in water and leached out during the solidifying step, creating enormous macro-voids inside the membrane at a higher loading of PEG. 

#### 3.2.1. Effect of Center-Tube Length on Morphology of PVC Fiber Membranes

This research investigates the different morphology by using polymeric hollow fibers which are prepared by the well-established dry/wet process using a tube-in-orifice spinneret. It can be expected that the morphology of the forming membrane results in dense skin on the inner surface which leads to the novel effect. According to the rapid liquid–liquid demixing between the solvent in the dope solution and water in the metal needle, the inner surface rapidly formed the solid phase faster than the outer surface.

As shown in Figure 4A, the SEM of the PVC hollow-fiber membranes prepared by a 0.3 cm metal needle length and controlled air gap of 1 cm, it was found that the effect of the metal needle length dramatically affects the morphology near the inner surface of the hollow fiber: finger-like structures were clearly seen with a 0.3 cm needle length, whereas the macro-void structure in the middle area became bigger than with the 1 cm needle length as shown in Figure 4B.

The mass transfer between the solvent in the dope solution and the non-solvent bore fluid in the metal needle occurred right away after the extrusion from the needle, acting akin to fast precipitation, which could be one explanation for the distinct morphology. Due to this, the 0.3 cm metal needle length showed more finger-like structures near the inner surface than the 1 cm case, which can be explained by Khayet [43]. He discovered that the hollow fiber was exchanged between the solvent and non-solvent at a lower air gap length than it could at a higher air gap distance. This can be compared to a shorter metal needle length (0.3 cm), which would result in a faster exchange of the solvent and non-solvent at the inner surface, possibly with less oriented polymeric chains and a higher free volume. To summarize the utilizing of the 0.3 cm metal length, this might be influenced by reducing the PEG chain mobility, PVC chain alignment, less oriented polymer chains, and high free volume. The current study observed that the increase and decrease in the metal needle length may result in significant effects on efficiency. Furthermore, these effects can be used to estimate the membrane’s efficiency by corroborating findings from other plausible studies. Ismail and colleagues [44] discovered that the finger-like structure within the structure provided a low wetting resistance and high permeability as a result of the PVC solution’s molecular chains tending to align themselves better and pack more closely to one another, approaching a dense skin layer. On the other hand, the sponge-like structure can be formed by the high viscosity of the spinning dope solution which leads to a decrease in mutual diffusion between the solvent and non-solvent, resulting in the slow solidification. To summarize the effect of the different metal needle lengths, if the non-solvent (water) in the bore fluid made contact with the dope solution before the outside precipitation, the skin formed on the inside surface of the fiber. The inner skin layer was precipitated immediately after extruding from the spinneret, whereas the outer skin layer was formed later when extruding into the coagulant bath, resulting in the difference morphology. In contrast, if the outside surfaces are reversed so that the bore fluid contains the non-solvent solution and the coagulation bath is water, the skin formed on the outside surface of the fiber is as shown in Figure 5 [10].

#### 3.2.2. Effect of Wind-Up Speed on Morphology of PVC Fiber Membranes

The effects of the wind-up speed (or take-up speed) and extrusion rates on the dimensions of the resulting hollow-fiber membrane is shown in Table 2. It was found that the outer and inner diameters decreased with the increase in the wind-up speed. This suggests that the morphology of the hollow-fiber membrane that was prepared from air gap 1 cm and metal needle length 0.3 cm become denser when the hollow fibers were obtained by the stress of the wind-up speed. In addition, using PEG as an additive resulted in an improvement in the precipitation rate of the spinning dope solution, resulting in a thin finger-like structure near the outer surface. Therefore, the increase in wind-up speed is directly feasible, increasing the solidification of the spinning dope solution. Moreover, this effect is referred to in previous studies [45,46], that it should be related to fabricating membranes with a macro-void free structure by the high drawing speed of the hollow-fiber membrane.

Three causes may be involved in the formation of the macro-void free structure which tends to occur in a suitable membrane for gas permeation: (1) the effect of the high drawing ratio may induce large negative normal stresses perpendicular to the skin surface and result in a radial outflow, consequently delaying non-solvent diffuse and macro-void agglomeration; and (2) the effect of high shear stresses resulting from smaller annulus gaps may provide oriented and denser polymer chain packing; thereby, the macro-void forming is hindered. When comparing the drawing ratios shown in Figure 6A–C, it can be seen that (1) high-speed drawing stabilizes the smooth skin, preventing the non-solvent intrusion from transforming into a polymer-rich phase and creating a macro-void free structure; and (2) the high solvent–coagulant miscibility causes the spinning dope solution to demix quickly, forming a structure resembling a finger [47].

The population of the very small pores and free volumes in the membrane’s structure was determined to grow considerably due to the random entanglement and alignment of polymer chains [48]. According to a report [44], the dope extrusion rate increases as it packs more tightly, improving and approaching a dense skin layer. Figure 6C shows the highest stretching; the chain molecules were oriented and in a more ordered arrangement. Therefore, the porosity diminishes as the drawing ratio increases and is further raised. Another reason for this effect is the evaporation time of the skin layer formation because it is expected that the DMF volatile solvent is removed from the fiber surface that results in a skin layer being formed during this dry phase-separated region [49]. For this reason, the work involving the DMF solvent from both the inner and outer surfaces causes the solvent from the 50 m/min wind-up speed to be withdrawn more quickly than the 40 and 30 m/min speed, which causes the solvent and non-solvent in the coagulant bath to exchange abruptly. At the same time, the solvent exchanging for the 40 m/min wind-up speed within the inner and outer surface is removed simultaneously, leading to the morphology of both sides being similar. In summary, a recent work showed the draw ratio data in Table 2 where the draw ratio increment can eliminate the macro-void structure during dry–wet spinning. To more obviously understand the increment of the wind-up speed or draw ratio, it might make polymer chains parallel and delay the insertion of an external coagulant; as a consequence, the cross-section morphology of the high wind-up speed showed the finger-like structure and dense skin on both the inner and external coagulant sides.

### 3.3. Contact Angle Measurement

The contact angle for membrane characterizations was found to be significantly less than 90°, suggesting that the membranes is more hydrophilic; the obtained data were reported in Table 3. A smoother surface and the highest hydrophilic property were noted to be shown on the PVC 580−G400 (15%) (80.76 ± 26). According to the presented data, it was found that the hydrophilic property can be determined by the increase in PEG contents; at the same time, comparing among PVC grades, it can be summarized that a low molecular weight showed more hydrophilic material than a high molecular weight that showed surface roughness in the SEM images. 

### 3.4. Dielectric Results 

It is well-known that molecular motions beneath the glass transition temperature $\left(T_{g}\right)$ appeared in sub-$T_{g}$ relaxation processes, as identified by the dielectric spectroscopy and dynamic mechanical analysis. The investigation of mechanical sub-$T_{g}$ relaxations in the separation application of membranes has paid particular attention to the influence of systematic changes in intra-segmental mobility and chain packing efficiency in permeability and selectivity [50].

According to the fact that PVC is a main key polymer and its dipole moment is perpendicular and rigidly attached to the main-chain backbone of the macromolecule, moreover, it is a strongly polar structure due to the existence of a C–Cl bond. During the course of this study, it became clear that the segmental dynamics and fractional free volume might be significantly impacted by the quite slight changes in the specifics of the PVC blended with the PEG molecular structure, which has a tendency to crystallize. Theatrically, during the drying process, the timescale of free volume relaxation may become larger than the timescale of diffusion [51], which is involved in the viscosity and the compatibility of the dope solution during the spinning process. The dielectric permittivity or dielectric storage $\left(\varepsilon^{\prime }\right)$ of the non-blended PEG and blended PEG were measured over the frequency range from 1 MHz up to 300 MHz at room temperature. The dielectric results are shown in Figure 7; it can be seen that the non-blended PEG membrane exhibits the highest values of permittivity at low frequencies. However, the dielectric permittivity of PVC-blended PEG-400 for 20 wt% was increased due to the phase separation of PEG from PVC during the demixing and spinning process. At the same time, some of the PEG contents covered the various contents and formed a hollow-fiber membrane that leads to the increased capacitance of the material rather than other PEG contents. Especially, the effect of PEG at a lower concentration (PEG 5 wt%, 10 wt%, and 15 wt%) lowered the dielectric constant because of the more compatible and miscible blending phase of PVC and PEG phase. Moreover, the different decreases in the dielectric constant in the lower frequency range of V14-G5, V14-G10, and V14-G15 are attributed to the decrease in contribution of both the interfacial polarization and conductivity.

### 3.5. Dynamic Mechanical and Thermal Properties

The effects of the blended PEG can be achieved and controlled by the incorporation of additives into the neat rigid PVC. External PEG—blended can change the PVC chain dynamic properties by providing free volume and by softening the intermolecular interactions which are involved in the viscoelastic relaxation properties, consequently, leading to a decrease in the glass transition temperature [52]. The PVC-blended PEG occurs through van der Waals and dipole–dipole interactions that can be established between the PVC polar carbon-chlorine bonds [53], and the PEG polar which consists of ether oxide units on the backbone structure; thus, most of these effects could be considered to observe variable properties of the blended membrane. This investigation is important in characterizing the material for its glass transition temperature. The different glass transition temperatures of the blended PEG were accessed by DMTA as presented in Figure 8. The changes in the storage modulus $\left(E^{\prime }\right)$ changed with the temperature as molecular motions within the polymer chains. The second-order transitions are observed by the remarkable drop in the storage modulus. The glassy-state modulus of the PVC-blended PEG−400 10 wt% is much higher than the other samples regardless of the temperature because it may well blend with PVC, resulting in the increase in $T_{g}$ being higher than the non-blended PEG and others. Due to the influence of PEG resulting in an increase in the viscosity which can obstruct segmental motions within the polymer-blended matrix, its effect can lead to the presence of a higher $T_{g}$, whereas, with higher PEG contents, the $T_{g}$ is raised. Thus, it can be assumed that the raising of $T_{g}$ influences the immiscibility between PVC and PEG. This effect indicates that the PEG has improved the softness of the membrane property which is involved in the increase in the amorphous state resulting in the increase in the gases’ permeability. The damping property as shown in Figure 9b $\left(\tan\delta \right)$ is the ratio of the dynamic loss modulus $\left(E^{\prime \prime }\right)$ to the dynamic storage modulus $\left(E^{\prime }\right)$, and it is clearly shown that the $T_{g}$ of the polymer matrix is found to decrease from 100 °C to 95 °C with the adding of PEG. It can be replied that the PEG miscibility blends with PVC and the blended PEG might be dispersed in the PVC matrix.

### 3.6. Membrane Flux Performance

A piece of the membrane with a known area was placed in a dead-end filtration unit by varying N_2_ gas pressures. A series of water flux (J) and its corresponding pressure (∆P) were recorded after each period of time. The hydraulic permeability coefficient (L_p_) was acquired from the graph slope amongst the flux and the pressure used: the relation was shown as J = L_p_∆P, known as the Hagen–Poiseuille equation [54,55,56]. The water flux (J) across the membrane was predicted and plotted against the used pressure. Figure 9 shows the water flux of the PVC 580 blending with PEG−400 membranes related to the PVC 580 pure and illustrates a linear relationship between the flux and the pressure used for the membranes. Table 4 shows hydraulic permeability (L_p_) values obtained from the slope of the graph in Figure 10, the PVC 580 pure and PVC 580 blending with 5% by wt. PEG−400 membranes were categorized as reverse osmosis, while the PVC 580 blending with 10% and 15% by wt. PEG−400 membranes were categorized as nanofiltration [57].

## 4. Conclusions

The preparation of a hollow fiber via a dry/wet spinning process is considered in terms of viscosity due to it being an important factor with which to adjust other parameters involving all procedures. The proposed operations are the appropriate hollow fibers for the required applications prior to determining the adjustable conditions. Preparing a suitable hollow-fiber membrane is feasible, especially when it uses PVC 580-blended PEG-400 5 weight percent with an air gap of 1 cm, a metal needle length of 0.3 cm, and a wind-up speed of 40 m/min. This is especially important when the flow configuration calls for countercurrent flow, where the feed flows in the fiber bore and the permeate is outside the fiber due to the fiber’s finger-like structures and dense inner surface. The addition of 5, 10, and 15% of PEG in PVC also affected the hydrophilicity and the morphology. When compared to the PVC 580 blending with 10% and 15%, PVC 580 pure, and PVC 580 blending with 5%, PEG−400 membranes demonstrated the lowest water flux. Therefore, this performance demonstrated the improvement of membranes for the appropriate water filtration.

## Figures and Tables

**Figure 1 polymers-16-02090-f001:**
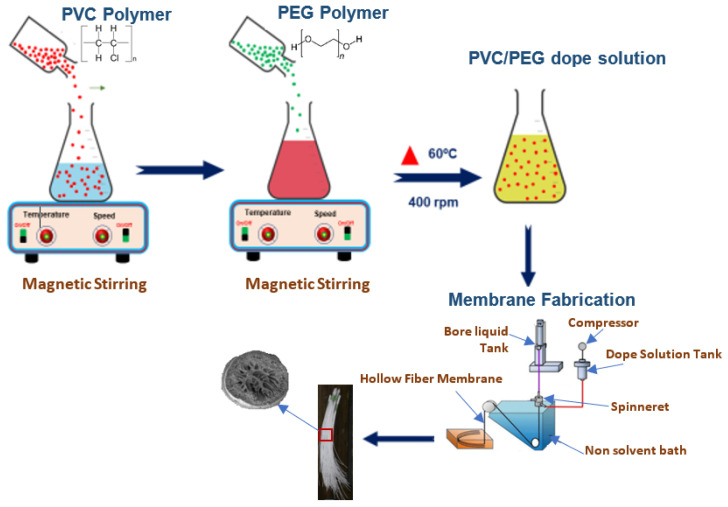
Fabrication of PVC/PEG hollow-fiber membrane.

**Figure 2 polymers-16-02090-f002:**
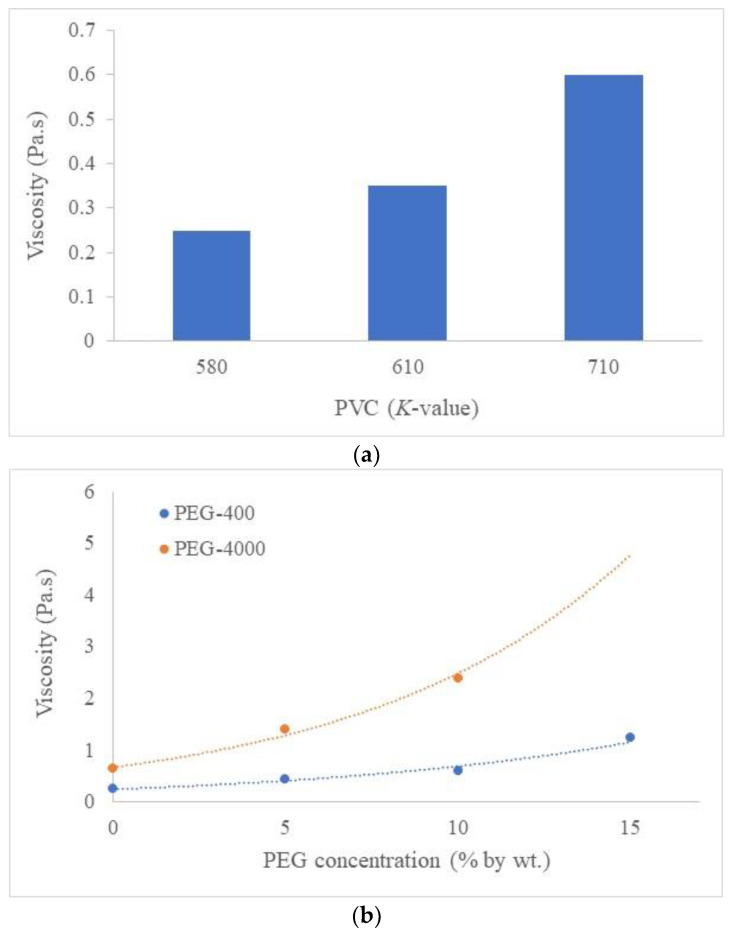
Viscosity results of (**a**) the different *K*-values of PVC, and (**b**) the different PEG-400 and PEG-4000 values.

**Figure 3 polymers-16-02090-f003:**
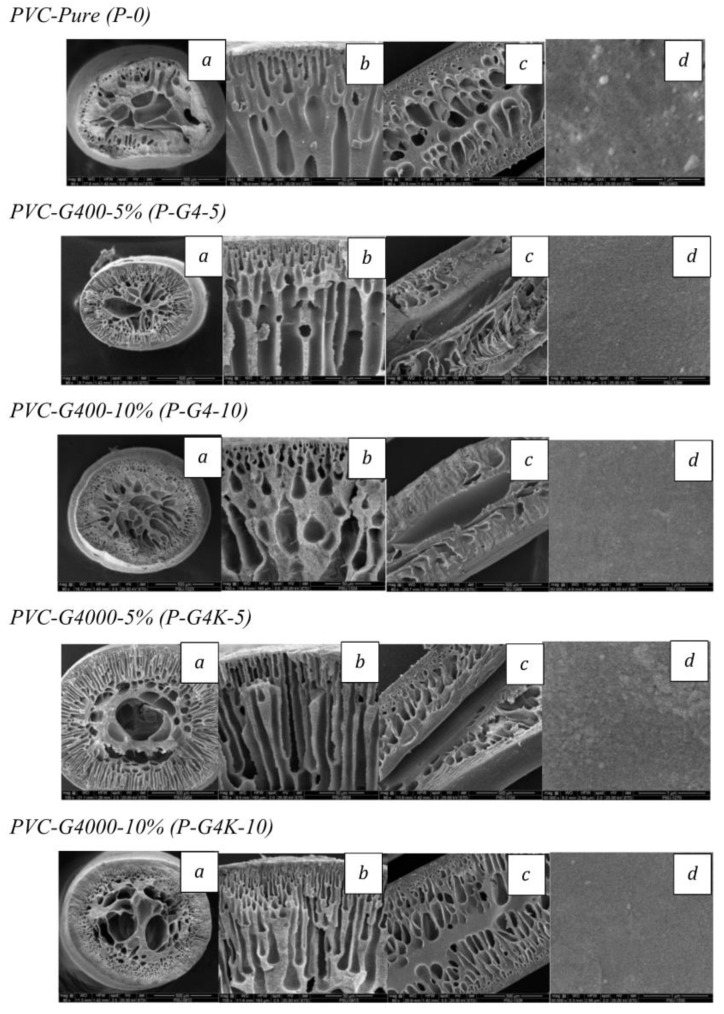
SEM images of PVC hollow-fiber membrane with unemployed center tube: (**a**) cross-section, (**b**) edge of cross-section, (**c**) cut-cleaved section, and (**d**) skin surface.

**Figure 4 polymers-16-02090-f004:**
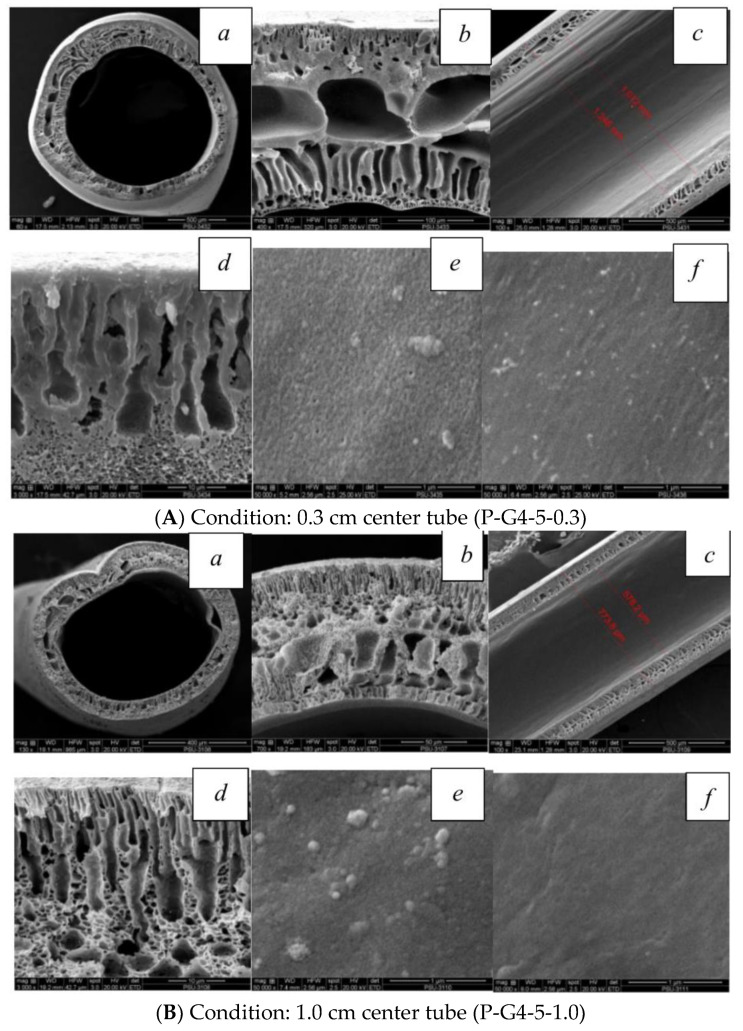
SEM images of PVC–HF membranes, wind-up speed of 30 m/min, with 0.3 cm (**A**) and 1 cm (**B**) center-tube length: (**a**) cross-section (**b**) edge of hollow-fiber membrane; (**c**) cut-cleaved section; (**d**) outer edge of outer layer; (**e**) outer surface; and (**f**) inner surface.

**Figure 5 polymers-16-02090-f005:**
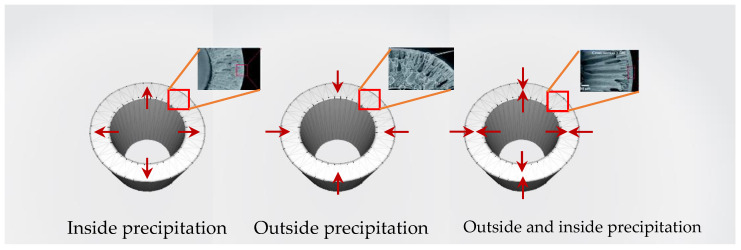
Depending on the bore fluid and the composition of the coagulation bath, the selective skin layer can be formed on the inside, the outside, or both sides of the hollow-fiber membrane [10].

**Figure 6 polymers-16-02090-f006:**
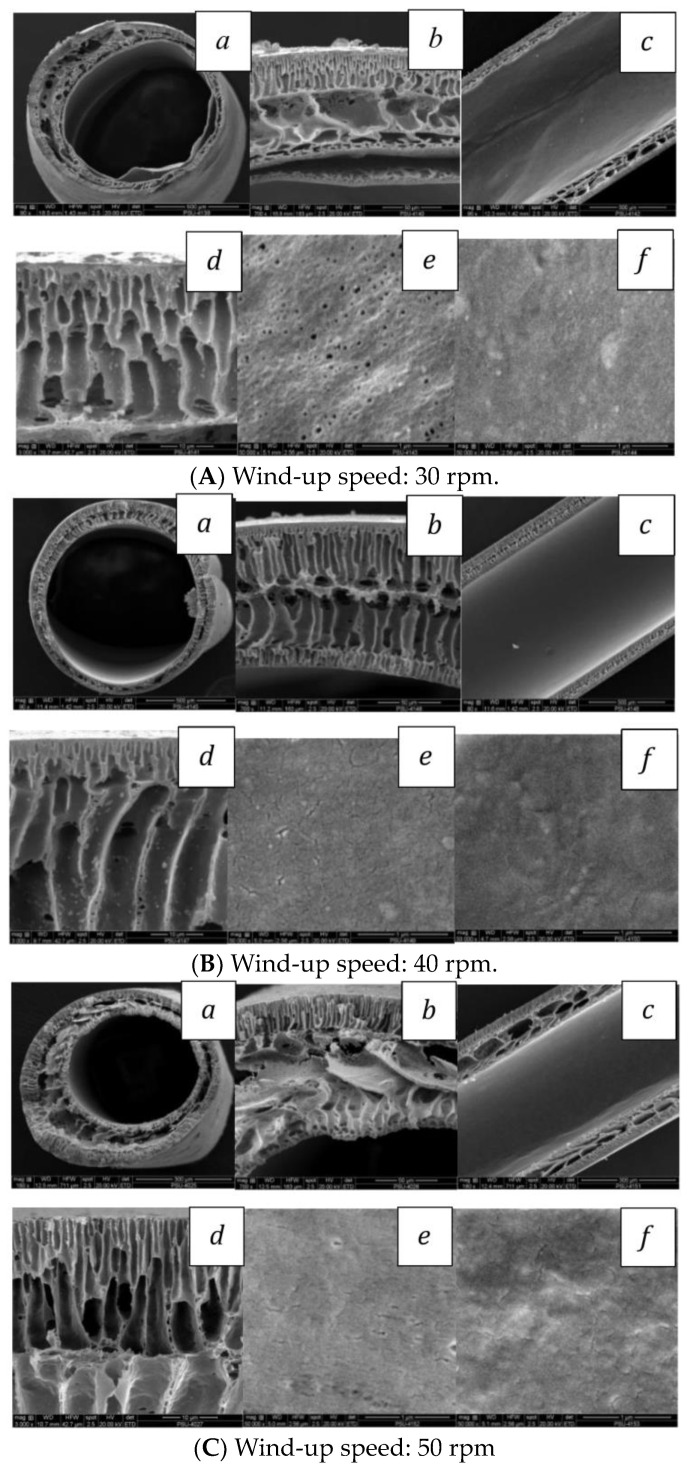
SEM images of PVC−G400-5% with 0.3 cm center-tube length: (**a**) cross-section; (**b**) edge of hollow-fiber membrane; (**c**) cut-cleaved section; (**d**) outer edge of outer layer; (**e**) outer surface; and (**f**) inner surface.

**Figure 7 polymers-16-02090-f007:**
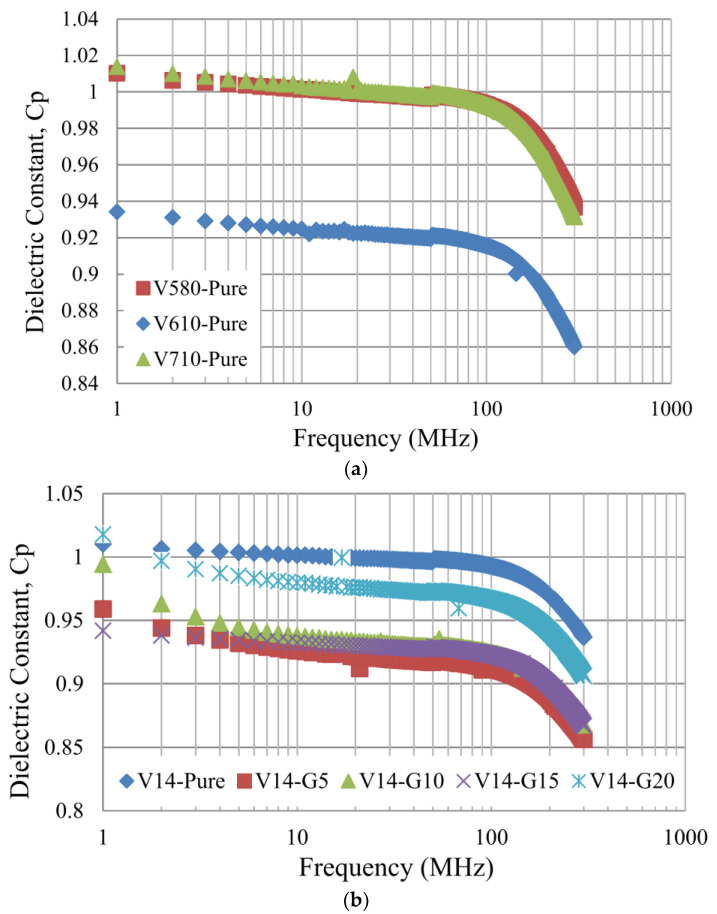
Dielectric property of (**a**) PCV—Pure and (**b**) PVC-blended PEG hollow-fiber membranes.

**Figure 8 polymers-16-02090-f008:**
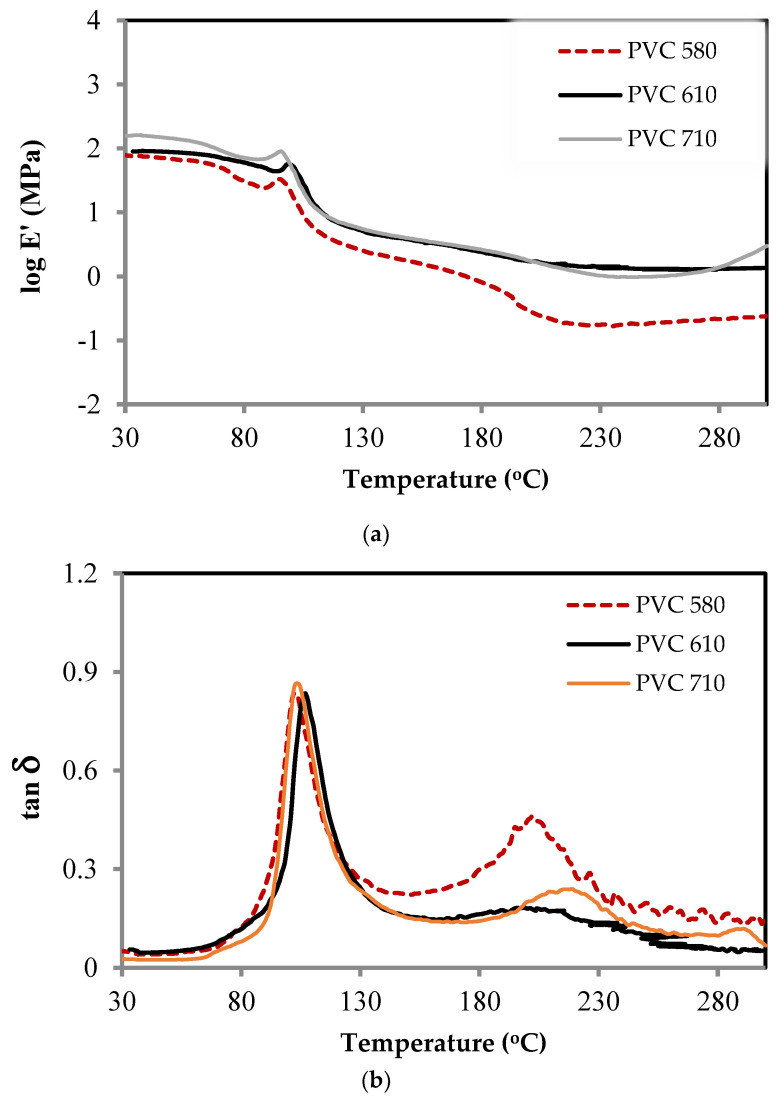
DMTA thermogram of PVC 580, 610, and 710: (**a**) storage modulus and (**b**) tan⁡δ.

**Figure 9 polymers-16-02090-f009:**
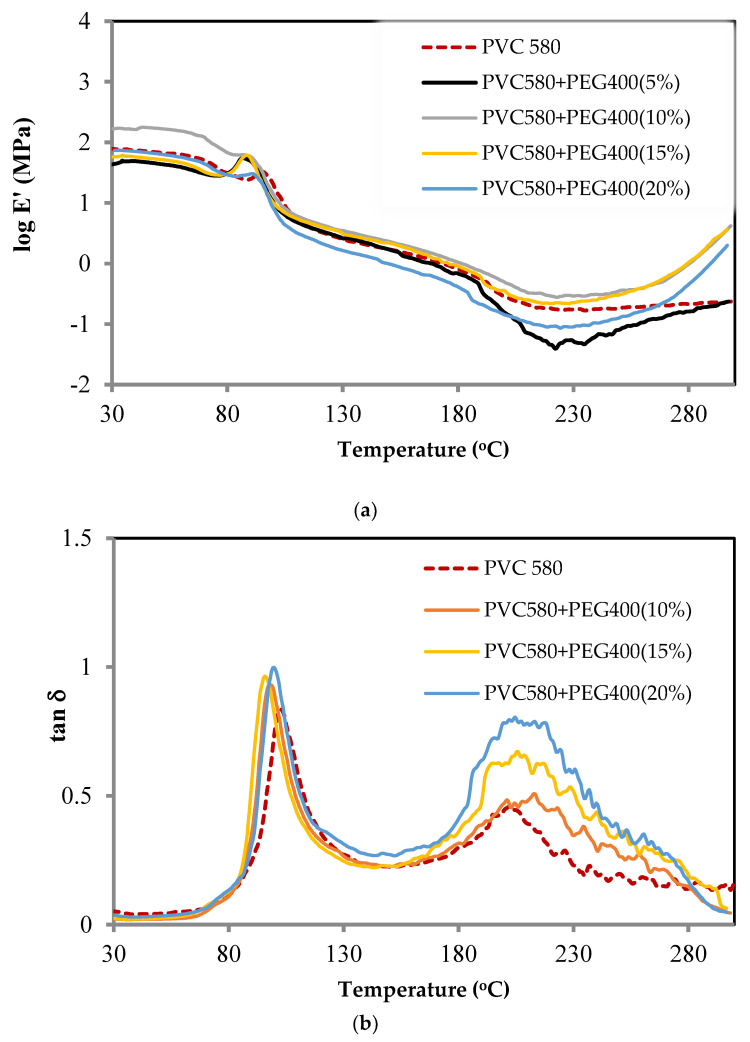
DMTA thermogram of PVC 580-blended PEG−400: (**a**) storage modulus and (**b**) tan⁡δ.

**Figure 10 polymers-16-02090-f010:**
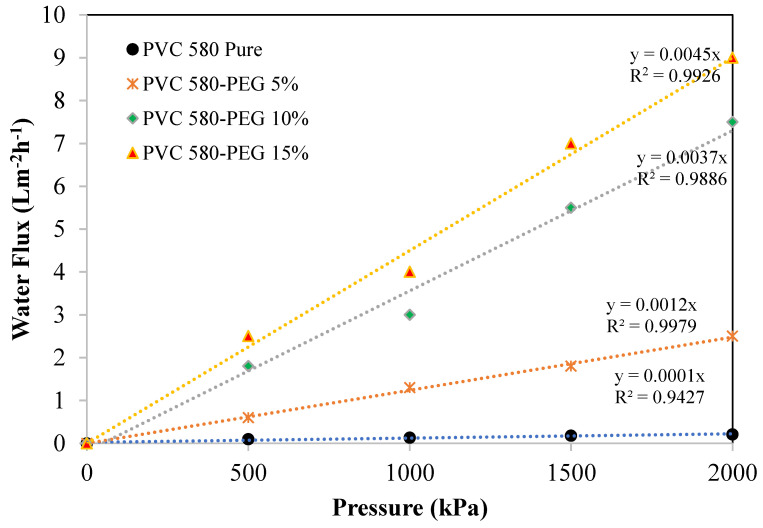
Water fluxes of the PCV—Pure and PVC-blended PEG hollow-fiber membranes.

**Table 1 polymers-16-02090-t001:** Spinning conditions of the fabricated PVC hollow-fiber membranes.

Parameters	Amount
Dope extrusion rate (mL/min)	2, 5, 10
Feed pressure (bar)	1
Bore fluid composition	Deionized water
Bore fluid flow rate (mL/min)	0.5
Center tube o.d. (cm)	0.1
Orifice o.d. (cm)	0.3
External coagulant	Deionized water
Air gap distance (cm)	0.5, 1.0, 1.5
Coagulant temperature (°C)	25
Room relative humidity (%)	60–70
Wind-up drum speed (rpm)	30, 40, 50

**Table 2 polymers-16-02090-t002:** The draw ratio on the different wind-up speeds of PVC–HF spinning *.

Sample	Extrusion Speed (m/min)	Wind-up Speed (rpm)	Draw Ratio (*λ*)	O.D. (µm)	I.D. (µm)	Thickness (µm)
A: P-G4-5	15	30	2.0	1200	800	350
B: P-G4-5	15	40	2.6	1100	850	220
C: P-G4-5	15	50	3.3	750	500	170

* Prepared from air gap 1 cm and center-tube length 0.3 cm.

**Table 3 polymers-16-02090-t003:** Contact angle results of flat sheet membranes.

Membrane	(*θ*)
PVC 580−Pure	88.28 ± 22
PVC 580−G400 (5%)	87.62 ± 32
PVC 580−G400 (10%)	88.34 ± 21
PVC 580−G400 (15%)	80.76 ± 26
PVC 610−Pure	89.36 ± 08
PVC 710−Pure	91.22 ± 63

**Table 4 polymers-16-02090-t004:** Hydraulic permeability of the membrane.

Sample	Lp × 10^−13^ (m^3^N^−1^S^−1^)
PVC 580 Pure	0.28
PVC 580−PEG 5%	3.32
PVC 580−PEG 10%	10.25
PVC 580−PEG 15%	12.47

## Data Availability

The data presented in this work are available upon request from the corresponding authors.

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
