# Peer review of "Effect of Viscosity and Air Gap within the Spinneret on the Morphology and Mechanical Properties of Hollow-Fiber Polymer Membranes for Separation Performance"

_polymers, 2024, doi:10.3390/polym16142090_

Round 1
Reviewer 1 Report
Comments and Suggestions for Authors
The process of forming filtration hollow-fiber PVC membranes is studied in the work. The material of the article is interesting for the scientific community and can be published. However, there are many comments on the work.
In the experimental part, it is necessary to provide a list of membranes with the composition of spinning solutions and formation conditions. Otherwise, it is difficult to follow the logic of the narrative.
Quantitative parameters of membrane properties should be given depending on one variable value: viscosity, molecular weight of the polymer or PEG concentration.
By what criteria were the membranes classified as reverse osmosis and nanofiltration? Salt rejection is not given. It is not difficult to calculate it if the electrical conductivity of feed, retentate and permeate are measured.
It is not clear how the thickness of the membrane calculated from the outer and inner diameter?
It is necessary to clearly formulate the novelty of the research and write the conclusions for the work, based on the results obtained. The conclusion do not show the effect of viscosity on the properties of membranes. The works that investigated the effect of spinning solution viscosity on the membrane properties are not cited.
It is not clear why the conclusion refers to gas separation membranes. The gas transport properties of the hollow fiber membranes have not been studied.
The work contains typos, the text is very difficult to perceive. It is necessary to carefully revise the manuscript.
Comments on the Quality of English LanguageThe work contains typos, the text is very difficult to perceive. It is necessary to carefully revise the manuscript.
Author Response
The process of forming filtration hollow-fiber PVC membranes is studied in the work. The material of the article is interesting for the scientific community and can be published. However, there are many comments on the work.
Comment 1: In the experimental part, it is necessary to provide a list of membranes with the composition of spinning solutions and formation conditions. Otherwise, it is difficult to follow the logic of the narrative.
Response 1: The manuscript used abbreviation instead of polymer composition as shown in Table 1. For spinning and formation condition is shown in manuscript (Table 1).
Table 1. Polymer composition and condition of hollow-fiber PVC membranes preparation.
|
Sample |
Composition |
Solution |
|
|
P-0 |
PVC (14% by wt.) |
|
DMF |
|
P-G4-5 |
PVC (14% by wt.) |
PEG 400 (5% by wt.) |
|
|
P-G4-10 |
PVC (14% by wt.) |
PEG 400 (10% by wt.) |
DMF |
|
P-G4-15 |
PVC (14% by wt.) |
PEG 400 (15% by wt.) |
DMF |
|
P-G4-20 |
PVC (14% by wt.) |
PEG 400 (20% by wt.) |
DMF |
|
P-G4K-5 |
PVC (14% by wt.) |
PEG 4000 (5% by wt.) |
DMF |
|
P-G4K-10 |
PVC (14% by wt.) |
PEG 4000 (10% by wt.) |
DMF |
|
P-G4K-15 |
PVC (14% by wt.) |
PEG 4000 (15% by wt.) |
DMF |
|
P-G4K-20 |
PVC (14% by wt.) |
PEG 4000 (20% by wt.) |
DMF |
Comment 2: Quantitative parameters of membrane properties should be given depending on one variable value: viscosity, molecular weight of the polymer or PEG concentration.
Response 2: As shown in Table 1 for quantitative parameters. The membrane properties were studied only the hollow-fiber PVC membranes formation because hollow-fiber was not formed in some polymer composition.
Comment 3: By what criteria were the membranes classified as reverse osmosis and nanofiltration? Salt rejection is not given. It is not difficult to calculate it if the electrical conductivity of feed, retentate and permeate are measured.
Response 3: This manuscript referred to previous studied of flat sheet membrane in ref. 15 and 16. For hollow-fiber PVC membranes was studied membrane flux as shown in Figure 10.
Comment 4: It is not clear how the thickness of the membrane calculated from the outer and inner diameter?
Response 4: The thickness of membrane was measured using SEM images.
Comment 5: It is necessary to clearly formulate the novelty of the research and write the conclusions for the work, based on the results obtained. The conclusion do not show the effect of viscosity on the properties of membranes. The works that investigated the effect of spinning solution viscosity on the membrane properties are not cited.
Respond 5: I added the effect of viscosity on the properties of membranes in conclusion part.
Comment 6: It is not clear why the conclusion refers to gas separation membranes. The gas transport properties of the hollow fiber membranes have not been studied.
Respond 6: The manuscript was referred to previous studied in ref. 15 and 16. The gas transport was studied with flat sheet membrane.
Comment 7: The work contains typos, the text is very difficult to perceive. It is necessary to carefully revise the manuscript.
Respond 7: I revied the manuscript with native speaker.
Reviewer 2 Report
Comments and Suggestions for Authors
The paper deals with the study of the effect of spinning parameters such as air gap and viscosity on the structure and performance of the hollow fiber membranes from PVC with the addition of PEG of different molecular weights. However, the experiment design and interpretation and discussion of the results don’t meet the criteria of high-quality publication.
There are several concerns and comments which the authors should address:
1. Line 19-20 “The various factors consisting of air gap, wind-up speed, dope extrusion rate, bore extrusion rate; besides, the different lengths of center tube which acted as inner-side fiber diameter during the preparation of hollow fiber membranes were studied”
However, according to table 1 air gap, dope extrusion rate, bore extrusion rate were constant in the study.
2. Keywords: PVC; PEG; Hollow Fiber Membranes; Nano-Cellulose; Wastewater
These keywords don’t fit to the paper: there is no any information that nano-cellulose was used. Moreover, there is no any information on the application of the prepared membranes for wastewater treatment.
3. Line 136: “K-value 580, 610 and 710”
There is no any decoding of K. What is it?
4. Line 141 “The resultant dope solution was continuously kept at 60 °C before spinning process”
Why? The comment is necessary in the text of the paper.
5. Line 155 Then mixed solution was poured into in-situ extruder which already installed with blunt needle.
It is not clear what is “in-situ extruder which already installed with blunt needle”.
6. Line 168-176; In the section 2 “Materials and Methods” section 2.3. Fabrication of PVC-PEG hollow fiber membrane the authors point out that “They were found that their selectivity and permeability were caused by the draw ratio and shear rate during spinning process. They described that elongation field stimulates the uncoiling motif of molecular chains; as a consequence, their compacted packing in the skin layer. In addition, they were found that the effect of draw ratio is affected to improving the membrane structure during spinning process due to the elongation arranged molecular orientation which accelerates polymer chains to provide more regular packing and more mono-disperse space. Besides, it is believed that the increasing air gap distance and wind-up speed may play an important role on membrane morphology due to the elongation stress is eliminated, retarded chain relaxation and orientation”.
This information is not related to Materials and Methods
7. In the section “2.3. Fabrication of PVC-PEG hollow fiber membrane” there is no description of the protocol for drying of membranes.
8. It is not clear what for the authors carried out dielectric measurements and what conclusions can be drawn from these data. For membranes from the pristine PVC and for membranes from PVC with the addition of PEG the values of dielectric constant in the range of frequencies of 1-10 MHz are 0.92-1.0. How significant is this difference? It would be more appropriate to evaluate the contact angle.
9. In the section 3.2. Morphology of PVC Fiber Membranes with Unemployed Center Tube and Controlled Air Gap 1.0 cm the results on the membrane spinning without bore fluid are shown. The idea of such experiments is not clear. These membranes are not hollow fiber membranes since the lumen (inner hole) is absent. These materials are more like rods.
10. The numbering of figures is not correct.
For instance
Line 267-268 “The cross section, cut-cleaved section, outer surface and skin layer were examined through SEM (Figure 2)”. However, on figure 2 the viscosity of the solutions is shown.
11.The authors state: ”As shown in Figure 3 (A), the SEM of the PVC hollow fiber membranes prepared by 0.3 cm metal needle length and controlled air gap of 1 cm, it was found that the effect of the metal needle length affects dramatically the morphology near the inner surface of hollow fiber: finger like structures were clearly seen with 0.3 cm needle length, whereas the macro-void structure in a middle area became bigger than with a 1 cm needle length as shown in Figure 3 (B)”
At Figure 3 SEM images of PVC fiber membrane with unemployed center tube are presented: (a) cross section, (b) edge of cross section, (c) cut-cleaved section and (d) skin surface. The differences of the membrane morphology depending on center-tube length are shown in Figure 4. However, according to the text at lines 292-293 “According to the mixing between solvent in dope solution and water in metal needle” the spinning of hollow fibers was carried out using bore fluid. Hence, the comparison of these structures is not appropriate.
12. Line 373-374:”With comparing the drawing ratio presented in Figure 5 (A, B and C)”
However, it is Figure 6.
13.Lines 343-344 “the skin will tend to be precipitated on the outside surface of the fiber as shown in Figure 7”
Apparently this means Figure 5., since Figure 7 shows the Dielectric property of (a) PCV-Pure..
14. Line 504: Figure 9 shows the water flux – It must be Figure 10…
15. In the Figures 3, 4 and 6 the magnification scale is practically indistinguishable.
16. Line: 358-361: “In addition, using PEG as additive resulted in an improvement in the precipitation rate of the spinning dope solution resulting in a thin finger-like structure near the outer surface. Therefore, the increase in wind-up speed is directly feasible increasing solidification of spinning dope solution.”
A very doubtful statement. An increase in stretching promotes the orientation of polymer chains and the formation of a denser structure.
17. Lines 386-391 The authors state “Another reason of this effect is the evaporation time of the skin layer formation because it is expected that the DMF volatile solvent will be removed from the fiber surface that results in a skin layer is formed during this dry phase-separated region. From this reason, due to this work involved with the DMF solvent from inner surface and outer surface; therefore, the solvent from 50 m/min wind-up speed is removed faster than 40 and 30 m/min resulting in the exchanging between solvent and non-solvent in coagulant bath occurs suddenly.”
It is a very doubtful statement. DMF is not a volatile solvent (boiling point is 153ËšC)).
18. In table 2 it is necessary to present the results of determining the transport characteristics of membranes.
19. In the section “3.5. Membrane Flux Performance” according to the measurements of water in a dead-end filtration mode the authors make the conclusion that “the PVC 580 pure and PVC 580 blending with 5% by wt. PEG-400 membranes were categorized as reverse osmosis while PVC 580 blending with 10% and 15% by wt. PEG-400 membranes were categorized as nanofiltration”. The similar conclusions are presented in the section 5. Conclusions.
The conclusion is unfounded. Experimental data on salt rejection are not presented.
Comments on the Quality of English LanguageModerate revision
Author Response
Comment 1: The paper deals with the study of the effect of spinning parameters such as air gap and viscosity on the structure and performance of the hollow fiber membranes from PVC with the addition of PEG of different molecular weights. However, the experiment design and interpretation and discussion of the results don’t meet the criteria of high-quality publication.
Respond 1: I would like to thank you very much for your comments. I will revise the manuscript according to suggestions. This work was continued from previous studies in Ref.15 and 16.
Comment 2: There are several concerns and comments which the authors should address:
Line 19-20 “The various factors consisting of air gap, wind-up speed, dope extrusion rate, bore extrusion rate; besides, the different lengths of center tube which acted as inner-side fiber diameter during the preparation of hollow fiber membranes were studied”
However, according to table 1 air gap, dope extrusion rate, bore extrusion rate were constant in the study.
Respond 2: The manuscript added all vary condition in Table 1. The manuscript selected the condition that the hollow fiber membrane formation.
Comment 3: Keywords: PVC; PEG; Hollow Fiber Membranes; Nano-Cellulose; Wastewater
These keywords don’t fit to the paper: there is no any information that nano-cellulose was used. Moreover, there is no any information on the application of the prepared membranes for wastewater treatment.
Respond 3: The manuscript was removed the keywords : Nano-Cellulose and Wastewater and was added the keyword : Spinneret.
Comment 4: Line 136: “K-value 580, 610 and 710”
There is no any decoding of K. What is it?
Respond 4: In this work used PVC resins which are classified by their K-value, an indicator of the molecular weight and degree of polymerization, were supplied by Thai Plastic and Chemicals Co., Ltd. The viscosity of PVC with different K-value were studied as shown in Figure 2.
Comment 5: Line 141 “The resultant dope solution was continuously kept at 60 °C before spinning process”
Why? The comment is necessary in the text of the paper.
Respond 5: The polymer solution and the mixtures were kept at 60oC to prevent the dissolution of PEG in the solvent.
Comment 6: Line 155 Then mixed solution was poured into in-situ extruder which already installed with blunt needle.
It is not clear what is “in-situ extruder which already installed with blunt needle”.
Respond 6: The polymer solution was extruded though spinneret. The spinneret was developed in the LAB as one part which is called “in-situ extruder”.
Comment 7: Line 168-176; In the section 2 “Materials and Methods” section 2.3. Fabrication of PVC-PEG hollow fiber membrane the authors point out that “They were found that their selectivity and permeability were caused by the draw ratio and shear rate during spinning process. They described that elongation field stimulates the uncoiling motif of molecular chains; as a consequence, their compacted packing in the skin layer. In addition, they were found that the effect of draw ratio is affected to improving the membrane structure during spinning process due to the elongation arranged molecular orientation which accelerates polymer chains to provide more regular packing and more mono-disperse space. Besides, it is believed that the increasing air gap distance and wind-up speed may play an important role on membrane morphology due to the elongation stress is eliminated, retarded chain relaxation and orientation”.
This information is not related to Materials and Methods
Respond 7: This information for “Fabrication of PVC-PEG hollow fiber membrane”. The manuscript gave more information on the effect of spinning conditions as shown in Table 1. The PVC with high K-value give higher viscosity of solution effect on formation of hollow fiber membrane.
Comment 8: In the section “2.3. Fabrication of PVC-PEG hollow fiber membrane” there is no description of the protocol for drying of membranes.
Respond 8: In this work, the hollow fiber membrane was dried at room temperature.
Comment 9: It is not clear what for the authors carried out dielectric measurements and what conclusions can be drawn from these data. For membranes from the pristine PVC and for membranes from PVC with the addition of PEG the values of dielectric constant in the range of frequencies of 1-10 MHz are 0.92-1.0. How significant is this difference? It would be more appropriate to evaluate the contact angle.
Respond 9: The manuscript was added results from contact angle as suggestion.
Comment 10: In the section 3.2. Morphology of PVC Fiber Membranes with Unemployed Center Tube and Controlled Air Gap 1.0 cm the results on the membrane spinning without bore fluid are shown. The idea of such experiments is not clear. These membranes are not hollow fiber membranes since the lumen (inner hole) is absent. These materials are more like rods.
Respond 10: In this work, the fabrication condition of hollow fiber membrane preparation was studied and found that Air Gap effects on hollow fiber formation of polymer. In this case, author select only condition which is optimization for hollow fiber membrane structure as requirement.
Comment 11: The numbering of figures is not correct.
For instance
Line 267-268 “The cross section, cut-cleaved section, outer surface and skin layer were examined through SEM (Figure 2)”. However, on figure 2 the viscosity of the solutions is shown.
Respond 11: I make correction the numbering of figures. Thank you very much.
Comment 12: The authors state: ”As shown in Figure 3 (A), the SEM of the PVC hollow fiber membranes prepared by 0.3 cm metal needle length and controlled air gap of 1 cm, it was found that the effect of the metal needle length affects dramatically the morphology near the inner surface of hollow fiber: finger like structures were clearly seen with 0.3 cm needle length, whereas the macro-void structure in a middle area became bigger than with a 1 cm needle length as shown in Figure 3 (B)”
At Figure 3 SEM images of PVC fiber membrane with unemployed center tube are presented: (a) cross section, (b) edge of cross section, (c) cut-cleaved section and (d) skin surface. The differences of the membrane morphology depending on center-tube length are shown in Figure 4. However, according to the text at lines 292-293 “According to the mixing between solvent in dope solution and water in metal needle” the spinning of hollow fibers was carried out using bore fluid. Hence, the comparison of these structures is not appropriate.
Respond 12: I make correction the numbering of figure from figure 3(A) and figure 3(B) to figure 4(A) and figure 4(B).
Comment 13: Line 373-374: ”With comparing the drawing ratio presented in Figure 5 (A, B and C)”
However, it is Figure 6.
Respond 13: I make correction the numbering of figure, Thank you very much.
Comment 14: Lines 343-344 “the skin will tend to be precipitated on the outside surface of the fiber as shown in Figure 7”
Apparently this means Figure 5., since Figure 7 shows the Dielectric property of (a) PCV-Pure..
Respond 14: I make correction the numbering of figure, Thank you very much.
Comment 15: Line 504: Figure 9 shows the water flux – It must be Figure 10…
Respond 15: I make correction the numbering of figure, Thank you very much.
Comment 16: In the Figures 3, 4 and 6 the magnification scale is practically indistinguishable.
Respond 16: I will used high resolution of SEM images in the figure 3, 4, and 6.
Comment 17: Line: 358-361: “In addition, using PEG as additive resulted in an improvement in the precipitation rate of the spinning dope solution resulting in a thin finger-like structure near the outer surface. Therefore, the increase in wind-up speed is directly feasible increasing solidification of spinning dope solution.”
A very doubtful statement. An increase in stretching promotes the orientation of polymer chains and the formation of a denser structure.
Respond 17: The morphology from SEM confirmed the formation of hollow fiber membrane porous with PEG as pore former related to Ref.45 and 46.
Comment 18: Lines 386-391 The authors state “Another reason of this effect is the evaporation time of the skin layer formation because it is expected that the DMF volatile solvent will be removed from the fiber surface that results in a skin layer is formed during this dry phase-separated region. From this reason, due to this work involved with the DMF solvent from inner surface and outer surface; therefore, the solvent from 50 m/min wind-up speed is removed faster than 40 and 30 m/min resulting in the exchanging between solvent and non-solvent in coagulant bath occurs suddenly.”
It is a very doubtful statement. DMF is not a volatile solvent (boiling point is 153ËšC)).
Respond 18: In this work, the hollow fiber membrane formation using phase inversion method, DMF will be removed and PVC form the hollow fiber membrane in water bath.
Comment 19: In table 2 it is necessary to present the results of determining the transport characteristics of membranes.
Respond 19: The hollow fiber membrane was selected for determining the transport characteristics as shown in Figure 10 and Table 3.
Comment 20: In the section “3.5. Membrane Flux Performance” according to the measurements of water in a dead-end filtration mode the authors make the conclusion that “the PVC 580 pure and PVC 580 blending with 5% by wt. PEG-400 membranes were categorized as reverse osmosis while PVC 580 blending with 10% and 15% by wt. PEG-400 membranes were categorized as nanofiltration”. The similar conclusions are presented in the section 5. Conclusions.
The conclusion is unfounded. Experimental data on salt rejection are not presented.
Respond 20: The membrane was classified with the slope of water flux to nanofiltration.
Reviewer 3 Report
Comments and Suggestions for Authors
Comments to the Author: The article describes the fabrication of hollow fiber membranes from the blending of Poly (ethylene glycol) (PEG) with Poly (vinyl chloride) (PVC) with different PEG molecular weights via dry/wet spinning process. The research has investigated the effects of air gap, wind-up speed, dope extrusion rate, bore extrusion rate and lengths of center tube on the preparation of hollow fiber membranes. In general, the manuscript is easy to follow. I only have some minor comments or suggestions, as follows:
1. For a better understanding of the spinning process, line 167, the unit of wind-up drum speed should be converted into “m/min.”
2. Line 268, “Figure 2” was wrong.
3. According to the SEM images in line 296, why were the macro-voids of the cut-cleaved section in P-G4K-10 smaller than those in P-G4K-5?
4. Lines 300 and 305 “Figure 3” was wrong.
5. Line 334, “Figure 7” was wrong.
6. Line 420, “Figure 6” was wrong.
Comments on the Quality of English LanguageComments to the Author: The article describes the fabrication of hollow fiber membranes from the blending of Poly (ethylene glycol) (PEG) with Poly (vinyl chloride) (PVC) with different PEG molecular weights via dry/wet spinning process. The research has investigated the effects of air gap, wind-up speed, dope extrusion rate, bore extrusion rate and lengths of center tube on the preparation of hollow fiber membranes. In general, the manuscript is easy to follow. I only have some minor comments or suggestions, as follows:
1. For a better understanding of the spinning process, line 167, the unit of wind-up drum speed should be converted into “m/min.”
2. Line 268, “Figure 2” was wrong.
3. According to the SEM images in line 296, why were the macro-voids of the cut-cleaved section in P-G4K-10 smaller than those in P-G4K-5?
4. Lines 300 and 305 “Figure 3” was wrong.
5. Line 334, “Figure 7” was wrong.
6. Line 420, “Figure 6” was wrong.
Author Response
Comments to the Author: The article describes the fabrication of hollow fiber membranes from the blending of Poly (ethylene glycol) (PEG) with Poly (vinyl chloride) (PVC) with different PEG molecular weights via dry/wet spinning process. The research has investigated the effects of air gap, wind-up speed, dope extrusion rate, bore extrusion rate and lengths of center tube on the preparation of hollow fiber membranes. In general, the manuscript is easy to follow. I only have some minor comments or suggestions, as follows:
Comment 1: For a better understanding of the spinning process, line 167, the unit of wind-up drum speed should be converted into “m/min.”
Respond 1: In this work, the wind-up drum speed use the rpm of motor (1 round = 1 m)
Comment 2: Line 268, “Figure 2” was wrong.
Respond 2: The numbering of figure was revised and correction.
Comment 3: According to the SEM images in line 296, why were the macro-voids of the cut-cleaved section in P-G4K-10 smaller than those in P-G4K-5?
Respond 3: The macro-void of P-G4K-10 is smaller than those in P-G4K-5 due to PEG is pore former, adding more PEG will increase pore density then the pore size is smaller than those of smaller amount of PEG.
Comment 4: Lines 300 and 305 “Figure 3” was wrong.
Respond 4: The numbering of figure was revised and correction.
Comment 5: Line 334, “Figure 7” was wrong.
Respond 5: The numbering of figure was revised and correction.
Comment 6: Line 420, “Figure 6” was wrong.
Respond 6: The numbering of figure was revised and correction.